# Microbiome Modulation as a Novel Strategy to Treat and Prevent Respiratory Infections

**DOI:** 10.3390/antibiotics11040474

**Published:** 2022-04-01

**Authors:** Barbara C. Mindt, Antonio DiGiandomenico

**Affiliations:** Discovery Microbiome, Vaccines and Immune Therapies, Biopharmaceuticals R&D, AstraZeneca, Gaithersburg, MD 20878, USA; barbara.mindt@astrazeneca.com

**Keywords:** bacterial and viral respiratory infections, microbiome modulation, gut-lung axis

## Abstract

Acute and chronic lower airway disease still represent a major cause of morbidity and mortality on a global scale. With the steady rise of multidrug-resistant respiratory pathogens, such as *Pseudomonas aeruginosa* and *Klebsiella pneumoniae*, we are rapidly approaching the advent of a post-antibiotic era. In addition, potentially detrimental novel variants of respiratory viruses continuously emerge with the most prominent recent example being severe acute respiratory syndrome coronavirus 2 (SARS-CoV-2). To this end, alternative preventive and therapeutic intervention strategies will be critical to combat airway infections in the future. Chronic respiratory diseases are associated with alterations in the lung and gut microbiome, which is thought to contribute to disease progression and increased susceptibility to infection with respiratory pathogens. In this review we will focus on how modulating and harnessing the microbiome may pose a novel strategy to prevent and treat pulmonary infections as well as chronic respiratory disease.

## 1. Introduction

Acute and chronic lower airway diseases are among the leading causes of death worldwide and the third most common cause of death in the United States. In 2021, infectious and chronic respiratory diseases accounted for more than 550,000 deaths in the US [1]. SARS-CoV-2 alone was the 3rd leading cause of death, outpacing all other causes with the exception of heart disease and malignant neoplasms [1]. Direct medical expenditures for the treatment of lower respiratory tract disease as well as indirect costs from the loss of productivity represent a considerable socioeconomic burden [2,3]. These conditions are becoming increasingly challenging to manage due to the continuous emergence of multi-drug resistant (MDR) bacteria and novel viral variants that promote infection and exacerbation of chronic lung disease. In the absence of effective countermeasures, it is estimated that infection with MDR strains will be responsible for 317,000 deaths each year in the US by 2050, a more than 10-fold increase to current numbers, underlining the urgent need for alternative prevention and intervention strategies [4,5]. The human body harbors trillions of microorganisms and comprises an intricate network of bacteria, archaea, protozoa and fungi as well as bacteriophages and eukaryotic viruses. It is estimated that the number of microorganisms that colonize the human body is at least equal to the number of somatic cells and that 500–1000 bacterial species inhabit our mucosal surfaces and the skin at any given time [6,7,8,9]. These microbes continuously interact with each other and with the human host, and this tightly regulated interplay is essential for the development and priming of the immune system and the maintenance of homeostasis [10,11,12]. When this symbiosis is perturbed due to alterations in the diversity and composition of the microbiome, the resulting dysbiosis can predispose to or exacerbate disease locally and at distal body sites. Dysbiosis can be induced by a variety of environmental factors such as treatment with antibiotics, diet and lifestyle or can be a consequence of chronic inflammation, infections or metabolic disorders [9,13,14,15,16,17,18,19]. Advances in culture-independent high-throughput bacterial sequencing approaches that enable large-scale taxonomic profiling of bacterial communities in patients and animal models, in combination with immunological and multi-omics datasets as well as mechanistic studies in germ-free rodent models, strongly suggest that aberrant microbiome alterations are a key driver of disease. A lot of these insights were gained in the course of the Human Microbiome Project (HMP), a 10-year interdisciplinary worldwide effort to characterize the composition of the healthy human microbiome at different body sites, analyze the relationship between microbiome changes and disease, and provide resources and novel technologies to broadly facilitate these studies in the scientific community [8,20,21,22]. Importantly, these studies also revealed a bidirectional cross-talk between the gut and lung microbiota, and growing experimental and clinical evidence show that changes in the intestinal microbiota can influence clinical outcomes of respiratory infections and chronic lung disease [23,24]. Since our understanding of other members of the human microbiome is still limited, we will briefly summarize implications of lung and intestinal commensal bacteria in lower respiratory tract infections and chronic lung disease and review how manipulation of the local and distal microbiota may aid in preventing disease while also serving as a source of novel drugs or drug targets for treatment.

## 2. The Lung and Gut Microbiome and Their Implications in Respiratory Disease

### 2.1. The Gut Microbiota

The gut microbiome performs many essential functions to maintain host homeostasis, including the breakdown of complex carbohydrates, synthesis of vitamins, maintenance of mucosal barrier integrity and protection against pathogens [25]. In addition, intestinal bacteria play an important role in the development and priming of local and systemic immunity [11]. The human gastrointestinal (GI) tract harbors more than 10^14^ bacteria and it is estimated that the genomic information contained in these bacteria outnumbers the genetic information contained in the human genome by at least 100-fold [26,27]. The healthy intestine contains >1000 bacterial species and despite a high degree of interindividual variability, can be reduced to a core microbiome that is dominated by taxa derived from the phyla Firmicutes, Bacteroidetes and to a lesser extent Proteobacteria, Actinobacteria and Verrucomicrobia [28,29,30,31,32]. As a consequence of oxygen availability, pH value and the presence of antimicrobial peptides (AMP) and bile acids, the composition and density of the microbiome varies greatly from the proximal to the distal GI tract. It gradually increases from ~10^2^ colony forming units/gram [CFU/g] luminal content in the acidic, sparsely populated bile acid- and AMP-rich environment of the proximal small intestine and reaches its highest density in the colon (~10^11^ CFU/g) [30]. Due to the higher availability of oxygen and the selective pressure provided by natural antimicrobials, the small intestine is mainly populated by fast-growing facultative anaerobic bacteria of the *Lactobacillaceae* and *Enterococcoceae* families, while the colon is dominated by fermentative anaerobes of the *Bacteroidaceae* and *Clostridia* families [30]. Colonization of the GI tract is induced during birth upon contact with the vaginal microbiota and once established remains relatively stable over time [9,30]. However, diet, lifestyle, and disease status can profoundly alter the microbiota and induce a dysbiotic state [13,14,15,16,19,33]. Microbiome-wide association studies (MWAS) in combination with mechanistic studies in germ-free or microbiota-depleted animals established the causal relationship between intestinal dysbiosis and the pathogenesis of a wide array of human diseases including not only gastrointestinal conditions but also systemic manifestations such as obesity, type 2 diabetes as well as allergic asthma and respiratory infections [34,35].

### 2.2. The Lung Microbiota

While it is well established that the upper respiratory tract (URT) of healthy individuals is continuously colonized by microbes, due to technical limitations, the lung environment was historically considered to be sterile. Recent advances in culture-independent bacterial sequencing approaches have now revealed that the lower respiratory tract (LRT) harbors a unique microbiome that plays a key role in promoting pulmonary homeostasis and may facilitate the priming of local immune cell populations [36]. In addition, the respiratory tract microbiome has emerged as a critical modulator of immune responses to respiratory infections and the pathogenesis of chronic lung disease [37,38,39,40]. Compared to the large intestine, the LRT colonization density in healthy individuals is relatively low, with 10^3^–10^5^ CFU/g lung tissue [41]. Under homeostatic conditions, the most abundant bacterial phyla in the human lung are Bacteroidetes (45–50%, genus *Prevotella*)*,* Firmicutes (30–35%, genera *Streptococcus*, *Veillonella*) and to a lesser extent Proteobacteria (10–15%, genera *Haemophilus, Neisseria*), Actinobacteria (5%, genus *Corynebacterium*) and Fusobacteria (5%, genus *Fusobacteria*) [41,42,43,44,45,46,47,48,49]. The lung microbiota is transient and maintained by continuous translocation of microorganisms from the URT and subsequent clearance by innate lung immune mechanisms, rather than local expansion of lung-resident bacteria [36,44,46]. This balance between migration and elimination is often perturbed during respiratory disease, which leads to overgrowth of bacteria with a competitive advantage and consequently a loss of microbial diversity. Structural alterations of the small airways and excessive mucus production are a hallmark of COPD, asthma and cystic fibrosis (CF). The resulting airway obstruction disrupts mucociliary clearance and the excess mucus may additionally support colonization with potential pathogens such as *Pseudomonas aeruginosa*, which is associated with increased mortality in CF and COPD [50,51,52,53,54,55]. In addition, defective clearance of bacteria by alveolar macrophages and/or airway neutrophils is observed in COPD [56,57,58,59,60], asthma [61,62], CF [63], idiopathic pulmonary fibrosis (IPF) [64] and after respiratory viral infections [65,66,67]. In accordance, patients suffering from chronic lung diseases exhibit elevated bacterial loads in their lungs, with increased abundance of potentially pathogenic proteobacteria including *Haemophilus*, *Moraxella* and *Pseudomonas* spp. [36,40,68,69,70]. Altogether, defects in mucociliary clearance as well as airway macrophage and neutrophil dysfunction may further contribute to disease progression and exacerbation by predisposing patients to bacterial infections.

### 2.3. The Gut-Lung Axis in Respiratory Disease

Increasing experimental and clinical evidence suggests that the gut microbiota and the lung are engaged in a continuous bidirectional cross-talk, termed the gut-lung axis and that dysbiosis at either site can contribute to the development and progression of distal diseases (Figure 1) [71,72]. Using animal models, it was shown that this dialogue is mainly facilitated by structural bacterial ligands, such as lipopolysaccharide (LPS), that bind and activate pattern recognition receptors (PRR) on host cells [38,73,74,75]. Microbiome-derived metabolites also play a critical role in this process [65,76,77,78,79]. In addition, the migration of activated immune cells from the intestine to the lung was described to aid in the defense against pulmonary helminth and bacterial infections [80,81]. Importantly, these gut-lung protective mechanisms could be directly stimulated by the intestinal microbiota and were abrogated in germ-free and antibiotic-treated mice, resulting in increased susceptibility to a wide array of bacterial and viral respiratory infections as well as chronic respiratory diseases [24,65,75,76,81,82,83,84,85,86].

Up to 50% of inflammatory bowel disease (IBD) patients and one third of individuals with irritable bowel syndrome (IBS) exhibit pulmonary manifestations ranging from subclinical aberrations to chronic lung disease [84,87]. The intestinal symptoms in IBD precede the lung phenotype, indicating that impaired lung function may be a consequence of a dysbiotic gut [88]. Furthermore, reduced intestinal microbiota diversity following antibiotic treatment during the first months of life as well as low levels of LPS and the microbiota-derived short-chain fatty acid (SCFA) metabolites are associated with the development of childhood asthma [89,90,91,92,93,94,95,96]. In accordance, genetically modified mice that are unable to inactivate LPS exhibit reduced pulmonary type 2 immune responses in a mouse model of allergic airway inflammation. This effect was ablated in mice treated with antibiotics and could be restored upon intrarectal administration of LPS, indicating a protective role of intestinal-derived LPS in allergic asthma [97]. Recent evidence suggests that SCFAs, the end products of dietary fiber fermentation by colonic commensals, are key modulators of the gut-lung axis and exert potent anti-inflammatory functions both locally and systemically. Oral supplementation with dietary fibers, SCFAs or SCFA-producing bacteria ameliorated pulmonary inflammation in mouse models of allergic asthma, further underlining the important role of the gut-lung axis in shaping lung immunity [76,78,86,98,99,100,101]. Cigarette smoke and aging are major risk factors for COPD development and are associated with shifts in gut microbial communities towards a decrease in alpha diversity as well as an increase in the Firmicute/Bacteroidetes ratio, established markers of intestinal dysbiosis [40,102,103,104]. Similar to allergic airway inflammation, intake of fermentable dietary fiber was associated with improved lung function in COPD patients and ameliorated lung pathology and inflammation in mouse models of COPD [105,106]. Furthermore, transfer of fecal matter from COPD patients into mice induced lung inflammation and accelerated the development of COPD-like pathology in respective animal models, further establishing a link of the gut microbiome to chronic lung disease etiology [107]. Importantly, the gut-lung axis is bidirectional, and it is well established that chronic lower airway diseases are often accompanied by manifestations in the gastrointestinal tract. For example, asthma patients exhibit structural changes in their intestinal mucosa, and COPD is often accompanied by chronic gastrointestinal tract diseases, intestinal microbiome alterations and lower levels of SCFAs compared to healthy controls [82,107,108,109,110,111]. However, the mechanisms behind these changes are only beginning to be understood and whether the differences in the gut microbiota are a cause and/or a consequence of chronic respiratory disease is often hard to discern and needs to be further addressed.

Besides chronic disease, intestinal dysbiosis was linked to the development of acute lower respiratory tract infections in children and evidence from mouse models show that the gut microbiota plays a key role in the defense against pulmonary pathogens [112]. Accordingly, germ-free and antibiotic-treated mice exhibit increased susceptibility to influenza virus [71,72,75,79] and respiratory syncytial virus (RSV) infections [83] and worse outcomes following lung infections with *Streptococcus pneumoniae* [38,67,74,81,113], *Klebsiella pneumoniae* [38,74,85], *Pseudomonas aeruginosa* [114,115] and *Staphylococcus aureus* [116]. Importantly, the administration of PRR agonists or the transfer of fecal material from control mice was sufficient to restore pulmonary immunocompetence and reversed susceptibility to lung infections in dysbiotic mice [38,72,74,75,85]. Furthermore, supplementation with dietary fibers and the metabolites SCFAs or desaminotyrosine (DAT) conferred protection against viral respiratory infections [79,83,117]. On the contrary, infections with influenza virus induce transient changes in the intestinal microbiota composition as well as gastroenteritis-like symptoms in mice and humans which cannot be explained by intestinal tropism of the virus [65,118,119,120,121,122,123,124,125]. These changes in the intestinal microbiota following influenza infection further predisposed the host to secondary enteric infections in mouse models of co-infection [121,124]. Influenza virus-mediated intestinal dysbiosis was also shown to feed back on the lung, leading to decreased bactericidal activity of alveolar macrophages and predisposing to pneumococcal pneumonia [89]. Importantly, alveolar macrophage function could be restored by oral SCFA supplementation [65]. Shifts in the intestinal microbiota towards increased abundance of Bacteroidetes and a decrease in Firmicutes were observed in a mouse model of RSV infection and specific microbial profiles were linked to disease severity in RSV-infected children [122,126]. Intestinal manifestations are also frequently reported in individuals infected with SARS-CoV-2, however, as opposed to influenza virus, there is clear evidence that SARS-CoV-2 is able to productively infect gastrointestinal epithelial cells and thereby directly causes intestinal dysbiosis [127,128].

## 3. Microbiome-Based Prevention and Intervention Strategies in Respiratory Disease

Given the implications of the intestinal microbiome in respiratory diseases, modulating the composition or activity of the resident microbiome by supplementation with bacterial substrates (prebiotics), live bacteria (probiotics, fecal microbial transplantation) or inanimate bacterial preparations (postbiotics) has been proposed as a novel prevention and intervention strategy for infectious and chronic lung disease. In addition, the selective depletion of opportunistic pathogens by antibacterial monoclonal antibodies or bacteriophages has shown promising results in preventing bacterial pneumonia in at-risk patients and both will be further discussed below and are summarized in Figure 2.

### 3.1. Supplementation with Prebiotics, Live Bacteria or Postbiotics

#### 3.1.1. Prebiotics

According to the International Scientific Association of Probiotics and Prebiotics (ISAPP), a prebiotic is “a substrate that is selectively utilized by host microorganisms to confer a health benefit to the host” [129]. Beneficial effects conferred by prebiotics are associated with modulation of the composition and/or activity of the commensal microbiota and include the expansion of beneficial bacterial taxa as well as the production of anti-inflammatory bacterial metabolites. The most well-studied and recognized prebiotics are non-digestible dietary fibers such as fructo-oligosaccharides, galacto-oligosaccharides, inulin and pectin, which are complex carbohydrates that are metabolized to SCFAs by commensal bacteria in the colon via anaerobic fermentation. Epidemiological and preclinical data strongly suggests a causative relationship between dietary fiber intake and lung health [130,131]. The generic Western diet contains low amounts of fermentable fiber and the associated gut microbiota harbors decreased microbial diversity, lower abundances of SCFA-generating bacteria and is associated with reduced lung function [132,133,134]. In accordance, increased dietary fiber intake positively correlated with pulmonary function and is associated with an up to 50% reduction in respiratory-related death rate [133,135,136,137]. It was suggested that the observed beneficial effects may be at least partly due to the biological functions of SCFAs such as acetate, propionate and butyrate. SCFAs exert important local physiological functions by stimulating intestinal epithelial cell turnover, promoting intestinal barrier integrity and serving as a vital energy source for colonocytes [138]. Furthermore, SCFAs exhibit anti-inflammatory and immunomodulatory functions either locally or at peripheral sites through activation of the G protein-coupled receptors (GPCRs) on epithelial cells and/or immune cells [139,140,141]. Several studies using mouse models showed that SCFAs promote the development and differentiation of extrathymic regulatory T cells (Tregs) [100,142,143,144] and can restrain allergic airway inflammation by impairing dendritic cells to drive T_H_2 effector functions and suppressing group 2 innate lymphoid cell (ILC2) activity [76,78,86]. In addition, SCFAs can promote lung immunity by modulating immune cell hematopoiesis in the bone marrow [78,115,117]. Immunomodulatory mechanisms of SCFAs in animal models are described in more detail in Table 1. Propionate and acetate are mainly produced by members of the Bacteroidetes, and Actinobacteria phyla such as *Bacteroides* spp. and *Bifidobacterium* spp., respectively, which can in addition to SCFAs, also generate lactate during dietary fiber fermentation [145,146]. The mucin-degrading bacteria *Akkermansia muciniphila* (Verrucomicrobia phylum) also produces both propionate and acetate [147] and Firmicutes, particularly *Faecalibacterium prausnitzii* (clostridial cluster IV), *Eubacterium rectale* and *Roseburia* spp. are the main butyrate-producing bacteria in the human intestine [146,148]. *Anaerostipes* spp. (clostridial cluster XIVa) and other colonic acetate- and lactate-utilizers can additionally generate butyrate through cross-feeding interactions [148]. Humans and mice consuming a diet high in fiber exhibited an elevated ratio of beneficial SCFA-producing Bacteroidetes to Firmicutes in the gut, which corresponded to higher local and systemic SCFA concentrations [78,100].

Consumption of a high-fiber diet or SCFA supplementation can ameliorate airway inflammation and improve lung function in asthmatic subjects and was protective in animal models of allergic airway inflammation [76,78,86,98,100,101,149,150]. Similar observations were made in COPD patients and animal models of COPD where high fiber intake was associated with a significant increase in lung function and decrease in COPD development and progression [105,106,151,152,153,154,155]. Other studies demonstrated that dietary fiber and SCFA supplementation was protective in experimental infection models of influenza and RSV as well as bacterial pneumonia and secondary bacterial infections following viral infections [83,117,156,157,158]. Direct evidence that the protective effects observed stem from SCFAs was obtained in studies using SCFA receptor-deficient mice which are more susceptible to bacterial pneumonia as well as respiratory viral infections, and exhibit exacerbated allergic airway inflammation in asthma models. However, the precise molecular mechanisms on how SCFAs mediate their beneficial effects are not yet fully understood. Dietary intake of fermentable fibers to promote the growth and activity of SCFA-producing gut bacteria and thereby remotely modulate lung immunity is a relatively easy and promising strategy to limit the onset and progression of chronic airway disease and may also confer protection against respiratory bacterial and viral infections. Clinical trials are underway to assess the potential effect of soluble dietary intake of inulin in COPD patients [159], and supplementation of asthmatics with a prebiotic that selectively promotes the growth and development of Bifidobacteria [160].

**Table 1 antibiotics-11-00474-t001:** Role of commensal-derived metabolites in mouse models of infections and chronic lung disease.

Metabolite	Disease Model	Immunomodulatory Effect	References
DAT	IAV	type I IFN signaling ↑ antiviral phagocyte response ↑	[79]
acetate	RSV	lung epithelial IFN-β ↑ antiviral activity ↑	[83]
acetate	IAV + *S. pneumoniae*	AM effector functions ↑ *S. pneumoniae* host defense following IAV infection ↑	[65]
acetate	*S. pneumoniae*	AM IL-1β and nitric oxide ↑ AM bactericidal activity ↑	[161]
acetate	*K. pneumoniae*	macrophage and neutrophil phagocytosis ↑ bacterial burden / lung inflammation ↓	[158]
butyrate	IAV	virus-specific CD8^+^ T cell response ↑ Ly6C^-^ monocyte hematopoiesis and differentiation to alternatively activated lung macrophages ↑ neutrophil response ↓	[117]
butyrate	heat exposure + IAV	lung IL-1β ↑ restores virus-specific CD8^+^ T cell and antibody responses upon heat exposure	[156]
butyrate	*K. pneumoniae*	HDAC ↓ IL-10 ↑ lung inflammation and pathology ↓	[157]
acetate	HDM	HDAC ↓ FoxP3 promoter acetylation and regulatory T cell function ↑ allergic airway disease ↓	[100]
butyrate acetate propionate	vancomycin + OVA, papain	lung DC migration to mLN ↓ antibiotic-induced exacerbated allergic response ↓	[76]
butyrate	OVA	eosinophil apoptosis ↑ lung eosinophil recruitment ↓ lung inflammation / eosinophilia ↓	[98]
propionate	HDM	DC precursor hematopoiesis ↑ seeding of lung with DCs exhibiting low Th2 polarization capacity ↑ allergic lung inflammation ↓	[78]
butyrate	IL-33	lung ILC2 function ↓ airway hyperreactivity ↓	[86]

Abbreviations: DAT, desaminotyrosine; IAV, influenza A virus; RSV, respiratory syncytial virus; HDAC, histone deacetylase; HDM, house dust mite; OVA, ovalbumin; DC, dendritic cell; mLN, mediastinal lymph node; ILC2, group 2 innate lymphoid cell.

#### 3.1.2. Fecal Microbiota Transplantation and Probiotics

Fecal microbiota transplantation (FMT) from healthy donors has been successfully used to treat patients with recurrent intestinal *Clostridioides difficile* infections and induce remission in ulcerative colitis patients [162]. These initial successes have sparked interest for the potential application of FMTs in extraintestinal diseases that are associated with gut dysbiosis including respiratory conditions. However, besides a proposed clinical trial administering FMTs to SARS-CoV-2 patients to potentially attenuate cytokine storm and pulmonary inflammation, the efficacy of FMTs for prevention or treatment of respiratory conditions has not yet been addressed in the clinic [163]. After fecal transplants contaminated with drug-resistant *Escherichia coli* resulted in the severe illness and death of clinical trial participants, safety concerns to enhance the donor screening process were voiced [164]. This incident and other safety issues have led efforts to focus on the development of defined and well-characterized formulations of fecal-derived live bacterial consortia or the use of known probiotic strains.

The ISAAP defines probiotics are “live microorganisms that, when administered in adequate amounts, confer a health benefit on the host” [165]. Probiotics are one of the most commonly consumed dietary supplements in the United States despite their efficacy being highly debated for many indications including respiratory diseases, and no probiotic has yet been approved as a live biotherapeutic agent [166]. However, favorable effects on the outcomes of respiratory infections and chronic airway disease were reported in some studies, mainly using animal models and in the context of upper respiratory tract infections in humans [167]. Due to their critical roles in immune maturation, their ability to generate bioactive metabolites and their critical role in the maintenance of intestinal barrier function and homeostasis, the most commonly used probiotics for preventive and therapeutic purposes are members of the *Bifidobacterium* (e.g., *B. breve*, *B. animalis*, *B. longum*) and *Lactobacillus* (e.g., *L. rhamnosus GG*, *L. paracasei*, *L. casei*, *L. plantarum*) genera, and to a lesser extent *Streptococcus thermophilus* and *Enterococcus faecium* [168]. Gut dysbiosis in early life characterized by reduced abundance and/or changes in in the composition of gut Bifidobacteria and Lactobacilli has been associated with allergic sensitization and the development of asthma [96,169]. Several studies suggest that preventive and therapeutic administration of probiotics may confer protection in animal models of allergic asthma. Mice that were supplemented with certain strains of Lactobacilli (e.g., *L. rhamnosus* GG, *L. reuteri*, *L. johnsonii*), Bifidobacteria (e.g., *B. lactis*, *B. breve*) or *Enterococcus faecalis* exhibited dampened allergic airway inflammation and airway hyperreactivity as well as decreased expression of pulmonary type 2 cytokines (IL-4, IL-5, IL-13), while anti-inflammatory cytokines such as TGF-β and regulatory T cells were induced [170,171,172,173,174,175]. Although these studies point to a potential protective role of probiotics, human studies testing the efficacy of supplementation with probiotics to prevent or treat asthma are so far inconclusive. Similar observations were made in cigarette smoke-induced COPD animal models where therapeutic and preventive dietary supplementation with *L. rhamnosus* or colonization with the commensal *Parabacteroides goldsteinii* ameliorated pulmonary inflammation and pathology [176,177]. However, there is no clinical evidence of a beneficial effect of probiotics on the onset or progression of COPD in humans.

The use of probiotics to prevent viral and bacterial lower respiratory tract infections yielded promising results in animal models. Oral supplementation or intranasal administration of the probiotics Lactobacilli or Bifidobacteria was protective in mouse models of influenza virus, RSV and mouse pneumonia virus infection [173,178,179,180]. Furthermore, dietary supplementation with *L. casei*, *L. rhamnosus* or *B. longum* enhanced lung clearance of *P. aeruginosa*, *S. pneumoniae* or *K. pneumoniae*, respectively, and ameliorated pulmonary inflammation [181,182,183]. Several clinical studies assessed whether dietary supplementation with probiotics leads to a reduction in the incidence and duration of acute respiratory tract infections. Meta-analyses showed a modest effect in reducing the duration and incidence of upper respiratory tract infections compared to placebo controls [184]. Efficacy of probiotics in specifically preventing lower respiratory tract infections has only been addressed in a few studies, including a clinical trial in high-risk ICU patients where daily oropharyngeal and gastric administration of *L. rhamnosus* significantly reduced the incidence of VAP compared to a placebo control [185]. In addition, COVID-19 patients orally receiving a multi-strain probiotic cocktail exhibited a decreased risk of developing respiratory failure compared to standard of care alone [186]. While these data are promising, further clinical evidence is needed to support the preventive or therapeutic use of probiotics in lower respiratory tract disease.

Besides the increasing the abundance of beneficial bacteria, dysbiosis can be counteracted by limiting the outgrowth of pathogenic bacteria. One potential approach to attack respiratory pathogens is the probiotic use of predatory bacteria from the genera *Bdellovibrio* spp. or *Micavibrio* spp. which specifically target and prey on other Gram-negative bacteria. *Bdellovibrio* attach to their prey, hydrolyze outer cell wall components and penetrate the periplasmic space, where they replicate and eventually burst the cell envelope of the host bacteria to start a new life cycle [187]. *Bdellovibrio* exhibit a broad host range, including major respiratory pathogens [115]. Importantly, *B. bacteriovorus*, the most extensively studied bacterial predator, was able to kill planktonic cultures as well as biofilm-embedded multidrug (MDR)- and extensively drug-resistant (XDR) clinical isolates of the opportunistic respiratory pathogens *K. pneumoniae, E. coli, Acinetobacter baumannii*, and *Pseudomonas aeruginosa* [188,189,190]. Another well-studied bacterial predator, *Micavibrio aeruginosavorus,* exhibits a more narrow host range than *B. bacteriovorus* and rather than invading its prey, *M. aeruginosavorus* attaches irreversibly to the prey cell surface and feeds on it while replicating externally, ultimately resulting in the death of the infected cells. *M. aeruginosavorus* shows potent in vitro killing activity when co-cultured with the respiratory pathogens *P. aeruginosa*, *K. pneumoniae* and *E. coli* [187]. Importantly, intranasal administration of both *B. bacteriovorus* and *M. aeruginosavorus* significantly reduced *K. pneumoniae* lung burden in a rat pneumonia model, had no adverse effect on the host in rodent models, and were cleared within days by innate immune mechanisms, encouraging their potential use to treat bacterial pneumonia in humans [191,192,193]. In addition, the administration of *B. bacteriovorus* generally exhibits low immunogenicity in vertebrate models due to the unique LPS composition, and no adverse effects were observed upon intravenous injection, ingestion or topical application in several independent studies [191,192,193,194,195]. Furthermore, *B. bacteriovorus* is present and abundant in the duodenum and ileum of healthy human individuals with significantly reduced abundance in Crohn’s disease and celiac disease patients, indicating a potential role of *B. bacteriovorus* in the maintenance of intestinal homeostasis [196]. While it is well established that predatory bacteria exhibit potent in vitro killing of Gram-negative respiratory pathogens, and no known adverse physiological effects were reported upon administration in animal models, further in vivo studies addressing safety concerns and effects on the host microbiome are needed. Predatory bacteria may also be for single use only due to the potential development of adaptive host immune responses.

#### 3.1.3. Postbiotics

Per ISAAP, postbiotics refer to “preparations of inanimate microorganisms and/or their components that confers health benefits in the host” that must stem from defined microorganisms [197]. Although they might be present in postbiotic preparations, substantially purified bacterial metabolites and molecules such as SCFAs, exopolysaccharides and proteins do not qualify as postbiotics due to the absence of cellular biomass [197]. The most commonly used postbiotics in the context of respiratory tract diseases are mixtures of whole and/or fractionated bacterial lysates of common respiratory pathogens which have been shown to reduce the frequency of acute recurrent respiratory infections [198,199,200,201,202]. Bacterial lysates were proposed to exert immunomodulatory functions by activating mucosal dendritic cells and stimulating mucosal pathogen-specific IgA responses, however the precise molecular mechanisms underlying their potentially beneficial clinical effects are not understood [203]. Polyvalent bacterial lysates such as Broncho-Vaxom or Ismigen are currently used for prophylactic purposes in the management of respiratory tract infections and COPD exacerbations. Modestly favorable outcomes were reported from a limited number of clinical trials. However, due to the relatively small sample sizes of these studies and contradictory results from similar studies, there is a clear need for more robust clinical trials to assess the preventive or therapeutic efficacy of these postbiotics.

### 3.2. Prevention and Treatment of Lower Respiratory Tract Infections by Selective Depletion of Opportunistic Bacterial Pathogens

Opportunistic respiratory pathogens including *P. aeruginosa*, *S. aureus*, *K. pneumoniae*, *S. pneumoniae*, *M. catarrhalis* and *H. influenzae* are commonly found in the human upper respiratory tract. While these bacteria are known to asymptomatically colonize healthy individuals, they can cause severe lower airway infections and exacerbations in patients suffering from chronic respiratory diseases. In addition, they are major etiological agents of hospital-acquired pneumonia (HAP) including ventilator-associated pneumonia (VAP), which due to their high incidence and mortality rates are a predominant cause of death among hospital infections. Treatment of these infections is further complicated by multi-drug resistant bacterial strains that are commonly isolated from the lower airways of HAP and VAP patients. Therefore, selective depletion of these bacteria in at-risk patient populations as well as therapeutic intervention by antibiotic-independent strategies to prevent and control potentially detrimental lung infections represents a critical area of research for the future management of multi-drug resistant lower airway infections.

#### 3.2.1. Antibacterial Monoclonal Antibodies

The development and use of monoclonal human antibodies (mAbs) that target and inactivate bacteria, their virulence factors and/or toxins is widely considered as one of the most promising antibiotic-independent approaches to combat infectious diseases [204]. Antibacterial mAbs exhibit several advantages over the use of conventional antibiotics. Due to their narrow target specificity, they exhibit no known adverse effects on the host microbiota and are less likely to induce widespread resistance [205]. Recent advances in human mAb technologies now allow for the design of polyvalent mAbs that can exert multiple mechanisms of action, including inactivation of virulence factors as well as complement the deposition and subsequent innate immune activation to further aid in bacterial clearance [206]. With a general half-life of several weeks, which can be further extended by introducing amino-acid substitutions in the Fc region to increase the binding to the neonatal Fc receptor, a single injection of a human mAb may be sufficient to provide protection against infection as opposed to multiple regimens of antibiotics a day [207,208,209]. While only three antibacterial antibodies targeting the exotoxins of *Clostridioides difficile*, *Clostridium botulinum* or *Bacillus anthracis* have been approved for clinical use so far, multiple mAbs for the management of lower respiratory tract infections are currently in clinical development [210,211]. *S. aureus* and *P. aeruginosa* are the leading causes of bacterial nosocomial pneumonia, including VAP, and are associated with significant mortality and morbidity [212,213,214]. Colonization of the upper airways with *S. aureus* or *P. aeruginosa* is a known risk factor for the development of VAP, and modulation of oropharyngeal colonization was shown to be effective in preventing VAP development [215,216]. Due to the high frequency of multi-drug resistant isolates, both pathogens are prime targets for the development of monoclonal antibodies to treat and prevent respiratory infections. Gremubamab (MEDI3902; AstraZeneca) is a bispecific human IgG1 mAb that selectively binds to the *P. aeruginosa* exopolysaccharide Psl and the type 3 secretion system (T3SS) protein PcrV, both highly conserved virulence factors, and was developed to prevent nosocomial *P. aeruginosa* pneumonia in high-risk patients [206,217,218]. Binding to Psl promotes complement fixation and opsonophagocytic bacterial killing by neutrophils as well as preventing the attachment of *P. aeruginosa* to airway epithelial cells whereas targeting PcrV inactivates the T3SS and enhances intracellular killing bacteria following phagocytosis [206,219]. Prophylactic as well as therapeutic administration of Gremubamab proved highly protective in rodent models of acute *P. aeruginosa* pneumonia as well as a rabbit model of VAP [206,220,221,222]. The effectiveness of MEDI3902 as a preventive measure for VAP in adult *P. aeruginosa*-colonized ICU patients was assessed in a phase 2 proof-of-concept study. Although the primary efficacy endpoint of reduction in *Pa* VAP incidence was not met in the total study population, risk reduction was observed in patients with lower baseline inflammation [223]. In addition to Gremubamab, the monoclonal human IgM antibody panobacumab (AR-101, Aerumab; Aridis Pharmaceuticals) is in clinical development for counteracting *P. aeruginosa* HAP. Panobacumab targets lipopolysaccharide from the highly prevalent *P. aeruginosa* serotype O11, which accounts for more than 20% of all nosocomial *P. aeruginosa* pneumonia cases [224]. Binding of Panobacumab to surface LPS results in complement-mediated clearance of *P. aeruginosa* by host phagocytes in vitro and the therapeutic administration of Panobacumab reduced bacterial burden and alleviated lung inflammation in a mouse model of acute *P. aeruginosa* lung infection [225,226]. Following promising results from a phase IIa study, Panobacumab is currently in late clinical development as an adjunctive therapy to standard of care antibiotics for hospital-acquired pneumonia [227].

Two antibodies are currently under clinical investigation for the prevention and treatment of nosocomial *S. aureus* pneumonia. Suvratoxumab (MEDI4893; AstraZeneca, outlicensed to Aridis Pharmaceuticals) specifically binds to and neutralizes the pore-forming alpha-toxin of *S. aureus*, a highly conserved key virulence factor which is expressed in a vast majority of clinical respiratory *S. aureus* isolates [228]. Suvratoxumab was shown to confer potent protection in animal models of lethal *S. aureus* pneumonia, and no severe adverse events were reported upon administration in healthy human adults in a phase 1 placebo-controlled trial [207,229,230]. Efficacy and safety of Suvratoxumab for the prophylaxis of *S. aureus* VAP in mechanically ventilated ICU patients colonized with *S. aureus* were further assessed in a phase 2 trial. While the study failed to meet its primary endpoint of a 50% reduction in *S. aureus* pneumonia in the Suvratoxumab versus the placebo control group, subgroup analyses of patients younger than 65 years showed significant reduction (47%) in the Suvratoxumab-treated group and an associated reduction in the duration of hospitalization and ICU stay [231,232]. A planned phase 3 study will further assess the efficacy of Suvratoxumab in VAP prevention in under 65 year old patients. Another human monoclonal antibody targeting *S. aureus* alpha-toxin, Tosatoxumab (AR-301, Salvecin; Aridis Pharmaceuticals), is currently in phase 3 clinical development for adjunctive therapeutic treatment of *S. aureus* VAP in combination with standard of care antibiotics [233]. In addition, several antibacterial human mAbs for the potential treatment of respiratory disease are in pre-clinical and early clinical development including AR-401 (Aridis Pharmaceuticals) and ASN-5 (Arsanis, outlicensed to BB200) targeting *A. baumannii* and *K. pneumoniae*, respectively [208].

#### 3.2.2. Bacteriophages and Phage Lysins

Bacteriophages are viruses that specifically infect bacteria by attaching to bacterial target cells, penetrating the bacterial cell membrane and injecting their genetic material into the host cytoplasm. Obligately lytic phages, which are predominantly used for phage-based therapeutics, then hijack the host’s transcriptional and translational machinery and replicate intracellularly. The newly-assembled virions mature and, after reaching a critical mass, phage-derived lytic enzymes (lysins), dissolve the bacterial cell wall to release the phage progeny into the environment [234]. Bacteriophages have been used as early as 1919 to treat bacterial dysentery but have been widely disregarded in Western medicine upon the discovery of antibiotics in the 1940s [234]. Meanwhile, phage-based therapeutics persisted in Eastern Europe and the former Soviet Union to treat, among others, respiratory infections with *S. aureus*, *P. aeruginosa, K. pneumoniae* and *E. coli* [234,235]. With the increase in drug-resistant bacterial isolates and the search for alternative intervention strategies, phage therapy has received renewed clinical interest in Western countries to treat respiratory infections with MDR bacterial pathogens. A clear advantage of using phages over antibiotics is their defined bacterial host range which allows them to selectively eradicate target bacteria without adversely affecting the host microbiota. In addition, their ability to penetrate and disrupt bacterial biofilms, as well as their compatibility with antibiotics and low immunogenicity renders them an attractive alternative to traditional antibiotics. Furthermore, phages have been shown to restore antibiotic-sensitivity to MDR bacteria and features of natural phages such as breadth of strain coverage and resistance development can be modified by genetic engineering to generate highly versatile synthetic phages [236]. Furthermore, nonlytic genetically engineered bacteriophages as a vehicle to selectively deliver genetic material encoding bactericidal proteins are currently in pre-clinical development by Phico Therapeutics to target *P. aeruginosa*-mediated VAP (SASPject PT3.9), *S. aureus* (SASPject PT1.2), *K. pneumonia* (SASPject PT4) and *E. coli* (SASPject PT5). Phage formulations exhibited potent efficacy in the treatment of *P. aeruginosa*, *K. pneumoniae*, *A. baumannii* or *E. coli* and in mouse models of lung infection and are currently being evaluated in the clinic for the treatment of human respiratory infections [237,238,239,240,241,242,243,244,245,246]. AP-PA01 (Armata Pharmaceuticals), a cocktail of four obligately lytic bacteriophages targeting *P. aeruginosa* respiratory infections, was used under the FDA expanded access program to successfully treat a CF patient suffering from an MDR *P. aeruginosa* infection as well as a patient with ventilator-associated pneumonia and empyema [247,248]. A second-generation phage cocktail, AP-PA02 (Armata Pharmaceuticals), with improved host range and increased potency, is currently being evaluated for safety, tolerability and preliminary efficacy in inhaled form in a Phase 1b/2a study (SWARM-Pa) in individuals with chronic *P. aeruginosa* lung infections and CF [249].

Besides using whole bacteriophage preparations, the use of phage-derived endolysins or engineered lysins was suggested to combat bacterial infections. Lysins exhibit several advantages over antibiotics and whole phage preparations, as resistance is unlikely to develop due to the conserved nature of their cell wall targets. Purified phage-derived and bioengineered chimeric endolysins show potent in vitro bactericidal activity against Gram-positive as well as Gram-negative respiratory pathogens including planktonic cultures and biofilms of *S. pneumoniae*, *S. aureus*, *P. aeruginosa* and *K. pneumoniae* [250]. Importantly, administration of phage endolysins drastically reduced *S. pneumoniae* titers in a mouse model of nasopharyngeal colonization and protected mice from fatal pneumococcal pneumonia or *P. aeruginosa* lung infection, respectively, underlining their potential to prevent and treat respiratory bacterial infections [177,251,252].

## 4. Conclusions and Future Outlook

Multi-drug resistant bacterial infections are on the rise and it is predicted that 10 million people will succumb to untreatable infections annually by 2050, estimated to surpass cancer and cardiovascular diseases combined [4]. In 2019 alone, 1.27 million deaths worldwide were directly attributable to infections with MDR pathogens, while 4.95 million deaths were associated with infections [253]. With 400,000 directly attributable deaths globally, lower respiratory infections are the leading cause of death among all MDR infections and are often associated with the priority pathogens *S. aureus*, *K. pneumoniae*, *S. pneumoniae*, *A. baumannii* and *P. aeruginosa*. Meanwhile, hundreds of millions of individuals globally suffer from chronic lung disease and respiratory viral infections that are becoming increasingly difficult to manage due to the emergence of novel viral variants. Hence, there is a critical need for novel therapeutic and preventive strategies, and due to the key role of lung and gut dysbiosis in lower airway disease, modulation of the microbiome has emerged as one potential intervention avenue.

Dietary supplementation with prebiotics, probiotics or postbiotics to deliberately alter the gut-lung axis would be a relatively simple approach for managing lower airway disease. Nevertheless, apart from a clear correlation of increased dietary fiber intake with lung health, there is limited evidence that intake of probiotics or postbiotics results in clinical improvement of lower respiratory conditions. This may at least in part be due to the high interindividual variability in response to colonization with probiotic strains where some individuals were shown to be more permissive, while others are resistant and host pathways were differentially affected upon colonization [254,255,256]. In addition, the effects of probiotics are generally transient, even in permissive individuals and only observed during or shortly after consumption [254,257,258,259,260,261]. The use of microbiome-specific tailored probiotics or purified microbial products and metabolites, such as SCFAs, could potentially overcome these issues. However, while SCFAs and SCFA receptor agonists were successfully used to treat and prevent lower respiratory infections and allergic airway inflammation in animal models, no efficacy was observed in the clinic so far [65,76,78,83,86,98,100,117,156,158,161].

Future research to better understand the complex host-microbiome crosstalk as well as interactions between individual members of the microbiota are needed and will aid in developing personalized strategies to modulate the microbiota in individual patients in order to achieve the highest possible treatment outcome. Promising results have been obtained from using antibacterial antibodies or bacteriophages in the prevention and treatment of nosocomial respiratory tract infections with opportunistic priority pathogens in at-risk patient populations. Further clinical evaluation is currently underway, and with innovations in phage and mAb technologies such as the generation of multivalent molecules with different modes of action and potential combined approaches of several mAbs to target multiple respiratory pathogens, they represent a viable and urgently needed alternative to current antibiotics [208,217].

## Figures and Tables

**Figure 1 antibiotics-11-00474-f001:**
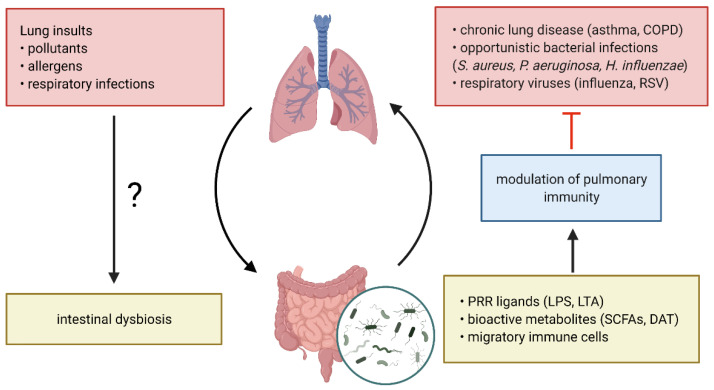
The gut-lung axis in respiratory disease. The continuous cross-talk between the gut and lung is facilitated by structural bacterial pattern recognition receptor (PRR) ligands including bacterial cell wall components, gut commensal-derived metabolites and migratory immune cells. Bacterial PRR ligands and metabolites are released in the circulation and bind to their respective receptors on pulmonary immune and/or epithelial cells, thereby modulating immunity to respiratory pathogens during chronic lung disease. Pulmonary insults can induce intestinal dysbiosis, however the underlying mechanisms are not well understood. COPD, chronic obstructive pulmonary disease; RSV, respiratory syncytial virus; PRR, pattern recognition receptor; LPS, lipopolysaccharide; LTA, lipoteichoic acid; SCFAs, short-chain fatty acids; DAT, desaminotyrosine. Figure created with BioRender.com.

**Figure 2 antibiotics-11-00474-f002:**
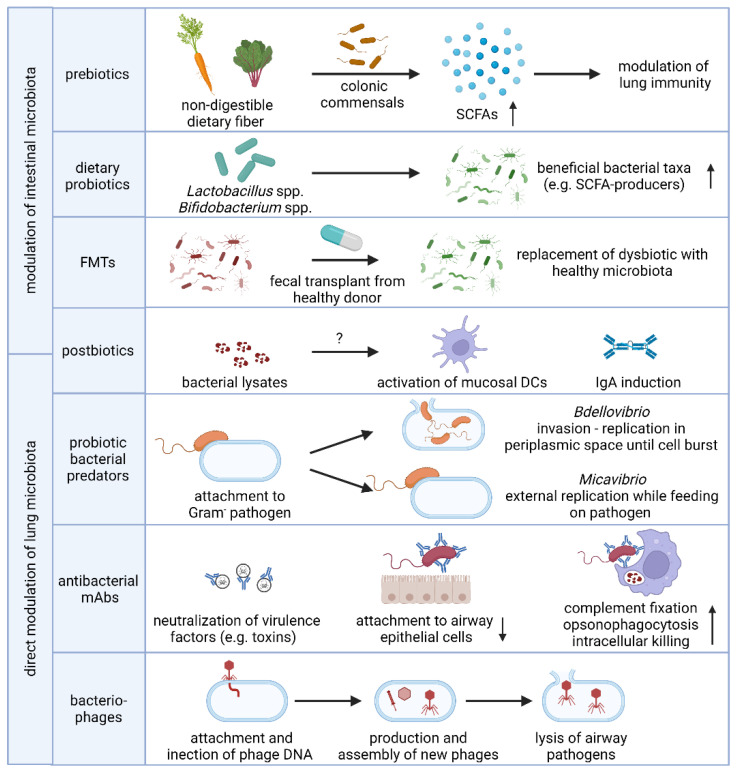
**Microbiome-based strategies to prevent and treat respiratory disease.** Modulation of the composition and/or activity of the intestinal microbiota to promote lung immunity aids in the defense against respiratory bacterial infections and management of chronic lung disease. Intake of non-digestible dietary fiber drives the production of the bioactive SCFA metabolites by colonic commensal bacteria while the use of probiotics aims to modify the composition of the microbiome towards beneficial bacterial taxa, including SCFA-producers. The precise immunomodulatory mechanism of intranasally or orally administered postbiotics remain elusive but may be in part due to mucosal DC and IgA induction. In addition, the lung microbiota can be directly modulated by selectively eliminating bacterial pathogens using probiotic predatory bacteria, antibacterial human mAbs or strain-specific bacteriophages. Abbreviations: SCFAs, short-chain fatty acids; FMT, fecal microbiota transplant; DC, dendritic cell; IgA, immunoglobulin A; mAb, monoclonal antibody. Figure created with BioRender.com.

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
