# Peer review of "Microbiome Modulation as a Novel Strategy to Treat and Prevent Respiratory Infections"

_antibiotics, 2022, doi:10.3390/antibiotics11040474_

Round 1

Reviewer 1 Report

The author summarized microbiome modulation as a novel strategy to treat and prevent respiratory infections. The manuscript is well written. Authors listed quite many applications on monoclonal antibodies and bacteriophages which is good.

- The author mentioned in the Abstract about SARS-CoV-2, but not explain more or stress on CoV-2 in the main body of the text. One should wither add more related results to the manuscript or reformulate the Abstract.

- line 488-489, ‘several antibacterial antibodies for the treatment of respiratory disease are in pre-clinical and early clinical development.’, is it possible to list some of them?

-Figure 1 is a bit simple, please improve.

- It is better to add a Figure 2 for the third section of the manuscript.

-Table 1 need to be optimized and compacted.

Reviewer 2 Report

The authors aimed at their work to summarize implications of lung and intestinal commensal bacteria in lower respiratory tract infections and chronic lung disease and review how manipulation of the local and distal microbiota may aid in preventing disease while also serving as a source of novel drugs or drug targets for treatment. Topic is of much interest due to the widespread of acute and chronic lower airway disease and overall work is interesting and very well written. I have some minor comments, if these corrections are made, I believe the above work can be published to antibiotics.

Make the review more appealing by adding additional figures and tables, such as to include Prebiotics and Postbiotics as an example.

Tables: improve the format to increase readability

Reviewer 3 Report

A manuscript entitled Microbiome modulation as a novel strategy to treat and prevent respiratory infections aims to review alternative preventive and therapeutic intervention strategies to combat airway infections in the future. 

The manuscript is well-written. Overall, the manuscript has provided good structure by good order of subheadings, such as in subheading The lung and gut microbiome and their implications in respiratory disease, i.e. gut microbiota, lung microbiota, gut-lung axis in respiratory disease.

Explanation on the Human microbiome project should be included in this article. This will help in providing a reason why aberrant microbiome alterations as the key driver of disease.

Minor comments:

Line 104-108: Lung microbiota is crucial to be explained in depth to provide an understanding what is the microbiota of healthy lung. Evidence from several studies will be more convincing to the readers. Authors are required to add deeper description of healthy lung microbiota. Perhaps, percentages, include figure of the microbiota composition will help to establish the understanding of healthy lung microbiota. Microbiota modulation in treating the lower respiratory infections will aim to regenerate healthy lung microbiota.

Explanation of Figure 1 is quite lengthy. It would be ideal to keep it short, and the detailed explanation extracted into new paragraph. For example, line 203 onwards can be a new paragraph. Lines 217-218 can be moved to 203. 
Line 214: pathogenic bacteria. Does this refer to the lung or to gut?

Lines 300-305. The statement in the sentences is risky without being supported by preliminary evidence. Perhaps it is best expressed as a suggestion. The success of FMT for Clostridioides difficile infections may not necessarily be the same case to attenuate cytokine storm and pulmonary inflammation. Animal model will be more suitable to test this potential application of FMT in the infections by SARS-CoV-2.

Lines 346-355. This is an important paragraph. It would be best to suggest "further evidence is required" at the end of this paragraph.

Line 356: Besides the increasing abundance.....

Line 358-387: bacterial predators.... this topic is in the section with FMT and probiotics subheading.
Please consider also having a figure in explaining the bacterial predator. 
Anyone has questioned whether bacterial predators are considered as probiotics?

Lines 425-435: These sentences need references.

Conclusions comprise some repetitive sentences. Authors are required to condense the conclusions further.
Lines 541-583: This paragraph is too long. Please condense and consider splitting the paragraph. 
Bacterial predator was not mentioned in the conclusion. If bacterial predators are considered probiotics, or not clear yet; authors are required to provide a short description within the subheading 3.1.2.
Reference 247 was used a few times as a single reference of key statements (Line 566, line 569). Authors are required to find similar articles supporting those statements.

Lines 569-574: Any reference(s) for these sentences?

Lines 579-583: Is there any reference for this?
